# Mechanical and Tribological Properties of Polytetrafluoroethylene Composites with Carbon Fiber and Layered Silicate Fillers

**DOI:** 10.3390/molecules24020224

**Published:** 2019-01-09

**Authors:** Andrey P. Vasilev, Tatyana S. Struchkova, Leonid A. Nikiforov, Aitalina A. Okhlopkova, Petr N. Grakovich, Ee Le Shim, Jin-Ho Cho

**Affiliations:** 1Chemistry Department, Institute of Natural Sciences, North-Eastern Federal University, Yakutsk 677000, Russia; gtvap@mail.ru (A.P.V.); sts_23@mail.ru (T.S.S.); nikiforov_l@outlook.com (L.A.N.); 2V.A. Belyi Metal-Polymer Research Institute of National Academy of Sciences of Belarus, Gomel 246050, Belarus; grapn@rambler.ru; 3School of Mechanical & Automotive Engineering, Halla University, Wonju 26404, Korea; elshim@halla.ac.kr; 4Department of Energy Science and Technology, Myongji University, Yongin 449-728, Korea

**Keywords:** polytetrafluoroethylene, polymer composite material, carbon fibres, layered silicates, synergetic effect, wear, friction

## Abstract

Mixtures of layered silicates (vermiculite and kaolinite) and carbon fibers were investigated as filler materials for polytetrafluoroethylene. The supramolecular structure and the tribological and mechanical properties of the resulting polymer composite materials were evaluated. The yield strength and compressive strength of the polymer increased by 55% and 60%, respectively, when a mixed filler was used, which was attributed to supramolecular reinforcement of the composites. In addition, the wear resistance increased by 850 times when using vermiculite/kaolinite fillers, which was due to protection of the surface by the formation of hard tribofilms.

## 1. Introduction

Polytetrafluoroethylene (PTFE) is commonly used as a matrix of composite materials for friction units in vehicles, because of its distinctive antifriction properties, and high chemical and frost resistances [1]. Pure PTFE has limited applications as an engineering material, due to its low wear resistance and cold flow. These limitations can be minimized by adding suitable fillers to a PTFE matrix to produce PTFE-based composites that are suitable for use as the friction units of technical equipment. The most common fillers used with PTFE are glass fiber, carbon fiber (CF), graphite, copper particle, molybdenum disulfide, and mixtures of these [2,3,4]. In particular, CF has been shown to significantly increase the wear resistance of PTFE at concentrations of less than 10 wt %, where the lowest wear rate was achieved with 10 wt % CF [5]. Wang et al. [6] also showed that PTFE composites with 10 wt % CF had the lowest wear rate.

In order to improve the mechanical and tribological properties of polymer composite materials (PCMs), layered silicates, including vermiculite (Vl) and kaolinite (Kl), are widely used as fillers [7,8]. Pre-treatment of layered silicates with surfactants is commonly performed, in order to improve compatibility between the polymer matrix and mineral fillers [7,8,9]. However, surfactants can easily be decomposed under the sintering temperatures used for PTFE (typically 360–380 °C) [9]. Therefore, the most suitable method for modifying layered silicates is mechanical activation [10,11]. It has been shown that mechanical activation enhances the compatibility between the polymer and layered silicates, by decreasing the filler particle size, and by increasing the structural activity of the filler. Sleptsova et al. demonstrated that the tribological behavior of PTFE can be improved by introduction of mechanically activated vermiculite (2–10 wt %), where the wear resistance of the polymer composite was 1700 times higher than that of the initial PTFE [10]. However, the mechanical properties of the polymer composites were significantly poorer than pure PTFE. A more significant improvement in the tribological behavior was obtained in their later study [12], in which PTFE was filled with layered silicates, such as serpentinite, phlogopite, and their mixtures; the mass wear rate increased by 2500 times compared to pure PTFE. However, the strength and elongation at break of the composites were lower than those of pure PTFE.

When combined fillers are added to a polymer matrix, each filler component has a different effect on the polymer matrix [13,14]. For example, the joint modification of polymers (epoxy and polyamide 66) with short carbon fibers (90 µm length) and nanoscale particles significantly improved the tribological properties of short-fiber-reinforced engineering polymers, especially at high friction load [15]. It was found that nanosized particles reduced adhesion interactions in the friction pair (counterbody composite), attributed to an increase in the distance between the steel counterbody and the composite material, i.e., the particles acted as spacers.

In this study, the mechanical and tribological characteristics of PTFE composites with mixed CF/layered silicate fillers were investigated in detail. The influence of the mixed filler on the formation of the composite structure was evaluated.

## 2. Results and Discussion

### 2.1. Mechanical Properties

Figure 1 compares the mechanical properties of PTFE and the various PCMs. As shown in Figure 1a, the tensile strengths of the PCMs were similar to that of the pure PTFE for CF concentrations of 1–8 wt %, regardless of the layered filler type. Further addition of CF up to 10 wt % resulted in a 20% reduction in the tensile strength, compared to that of the pure PTFE. The elongation-at-break values of the PCMs were similar to that of the initial PTFE over the entire CF concentration range, independent of the layered filler type (Figure 1b). As shown in Figure 1c,d, the addition of the mixed filler increased the compressive stress and yield stress, respectively, by about 50% compared to the initial polymer matrix at CF contents of 8–10 wt %. This indicates that the mixed fillers significantly reinforced the PTFE polymer matrix. In general, our results showed that the mechanical properties of the PCMs were dependent on CF concentration, but not on the filler type.

PTFE is a crystallizable polymer, and its phase composition and supramolecular structure depend on the filler when it is used as a matrix in a PCM [16]. These factors determine the properties of the resulting composites. Therefore, the structures of PTFE and its composites were investigated using scanning electron microscopy (SEM) (Figure 2). It can be seen that the supramolecular structure of the PCMs differed from that of the initial PTFE, which was characterized by a lamellar structure, as shown in Figure 2a. However, a comparison of the supramolecular structure of the PCMs with the CF and layered silicate filler mixtures (Figure 2b,c) showed that the structure was not dependent on the silicate filler type. The CFs were uniformly distributed and randomly oriented in the PCM. A complex surface with small structural elements was observed for the PCMs. Such changes in the supramolecular structure with the addition of the combined filler resulted in a significant increase in the degree of crystallinity, and a decrease in the density of the PCM, with increasing CF concentration (see Table 1).

Table 1 shows the theoretical and experimental density values of the PCMs, along with their degree of crystallinity, as determined by XRD. The density of the initial PTFE was 2.16 g/cm^3^, with a degree of crystallinity of 63%. The difference between the theoretical and experimental density of PTFE and PCM was calculated using the following equation [17]:(1)$$\rho_{add}=\frac{1}{\frac{K\left(1-\varphi \right)}{\rho_{c}}+\frac{\left(1-K\right)\cdot \left(1-\varphi \right)}{\rho_{a}}+\frac{\varphi }{\rho_{f}}}$$ where *ρ**_f_* is the filler density, *ρ**_c_* is the extrapolated density of the crystalline phase (~2.30 g/cm^3^), *ρ**_a_* is the extrapolated density of the amorphous phase (~2.04 g/cm^3^), K is the degree of crystallinity (%) determined by XRD, and *φ* is the filler weight content (wt %) [17]. Table 1 shows that the addition of the combined fillers resulted in an increase in the degree of crystallinity from 63% for the initial PTFE to 74% for PCM, regardless of the layered filler type. The density of the PCM decreased proportionally with increasing total filler content, which is consistent with the low density of CF (~1.45 g/cm^3^). The significant difference between the theoretical and experimental PCM densities was explained by the low density of the CF, and also a “loosening” of the supramolecular structure in the PCM.

The mechanical properties of PCMs with fiber fillers depend on both the adhesion between the components and the properties of the interfacial layer at the polymer–fiber interface [18]. The surface of the CFs used here were modified using a plasma–chemical treatment with fluororganic compounds, which reduced the surface energy of the fibers, resulting in better wetting of the fiber surface with the PTFE, thereby, increasing their compatibility [19]. Figure 3 shows an SEM image of the PCM supramolecular structure on the fiber surfaces. It can be seen that the formation of PTFE microfibrils on the surface of the CFs was not dependent on the presence of the layered silicates. Similar PTFE microfibril fiber surfaces were observed previously for CFs modified by oxidation with air and ozone [20]. The interfacial region of the PTFE and CF showed a similar structure to samples prepared in various previous studies [20,21,22]. Figure 3b shows that the PTFE microfibrils formed a dendritic structure on the surface of the CFs. 

Interactions between the polymer and CF occurred locally (not over the entire CF surface), due to the adhesion between separate PTFE microfibrils and the CFs. Therefore, dendritic structures formed on the CF surface. These structures were characterized by an exfoliated supramolecular structure on the interface boundary. This structure could have contributed to the measured density being lower than the theoretical one; this structure clearly had a lower density than the other samples. A schematic diagram of the structure of the PTFE and its interactions with the CFs is shown in Figure 4.

### 2.2. Tribological Properties

PTFE has a low friction coefficient and a relatively low wear resistance, due to its molecular and supramolecular structure [23]. However, as shown in Figure 5, the tribological properties of PTFE changed when the mixed fillers were added; specifically, a sharp increase in the wear resistance compared to the pure polymer. As shown in Figure 5a, the wear resistance of the PCMs with mixed fillers increased approximately linearly up to 5 wt % CF, and then it was constant for compositions with 5–8 wt % CF. This trend was independent of the type of layered silicate, probably due to their very low content compared to that of the CF. The wear rate of the PCMs decreased by a factor of ~750 compared with that of pure PTFE. As shown in Figure 5b, the friction coefficient of the PCMs increased by 20% compared with that of pure PTFE. Composites containing 1 wt % mechanically activated Kl and 1–8 wt % CF showed the lowest friction coefficients, which is desirable for their use as friction parts. The friction coefficients of composites with 1 wt % Vl and 1–8 wt % CF were 36% higher than that of the initial polymer, while a further increase in CF content up to 10 wt % resulted in a further increase in the friction coefficient of the PCM.

The worn surfaces of pure PTFE and the PCMs were investigated using SEM, as shown in Figure 6. The worn surface of the pure PTFE (Figure 6a) showed grooves along the sliding direction. Figure 6b,c show that the worn surfaces of the PCMs with combined fillers were very similar, where the CFs were distributed randomly and protruded from the worn surfaces, protecting the underlying polymer from abrasion. The amount of fibers visible on the worn surfaces increased with increasing CF content in the PCM. 

The FT-IR spectra of the PTFE and PCM specimens are presented in Figure 7. There was no significant difference between the IR spectra of the PTFE before and after friction. For all of the PCM samples (regardless of the layered silicate type or CF content), new absorption bands were observed after wear tests in the regions 3600–3200 cm^−1^, 1650–1654 cm^−1^, and 1428–1432 cm^−1^. These new peaks appeared due to oxidation processes on the surface of the polymer under friction [24]. 

In a previous study, the decreased wear rate of PTFE/α-Al_2_O_3_ composites compared to pure PTFE was attributed to tribochemical reactions, which resulted in wear debris with different physicochemical properties [24,25]. The formation of a strong transfer film on the surface of the composite can increase wear resistance. Similar tribochemical reactions can take place during the friction of the composites, as Vl and Kl consist of oxides and silicates. This is thought to be the reason for the observed changes in surface morphology and composition, as shown in Figure 6 and Figure 7, respectively.

In order to determine the effect of the layered silicates on the tribological properties, the wear rates of PCM with mixed fillers and only CF were compared, as shown in Figure 8. The introduction of 1 wt % mechanically activated layered silicates and 5–8 wt % CF to PTFE resulted in wear resistance values that were 2.5–3 times higher than that of the composites containing only of CF filler. The tensile strength and elongation at break of these PCMs were similar to those of the initial polymer.

SEM images of worn PCM surfaces are shown in Figure 9, where the morphology of the PTFE/CF interface differed for PCMs with a combined filler or only a CF filler. The arrows in Figure 9a,c indicate the detachment of CFs from the friction surface. Protruding CFs on the PCM surface protect the worn surface from fracture during the friction process [26]. Areas with high roughness experience cyclic impact from the counterbody, leading to a local increase in the load on the fibers at the surface [27,28]. As a result, the CF were abraded and removed from the friction surface of the PCM, as shown in Figure 9c. The SEM images in Figure 9b,d demonstrate that the adhesion between the CF and matrix was much better when layered silicates were also added as filler. This can be explained by the formation of secondary reinforcement in the form of tribofilms between fibers on the worn surface [29]. The formation of such tribofilms was supported by the tribochemically degraded PTFE detected by FT-IR spectroscopy (Figure 7), and the change in the morphology of the worn surface of the composites with mixed fillers, compared to that of composites with only CF (Figure 10). The surface of the PCM with only CF as filler (Figure 10a) was smoother than those of the PCMs with mixed fillers (Figure 10b,c). This was due to the formation of the fragmented tribofilms between adjacent fibers on the surface of the composites with the mixed fillers. These tribofilms were thought to contribute to the increase in wear resistance of the PCM with layered silicate fillers. The formation of tribofilms can decrease load transfer on CFs during friction, and lead to improved wear resistance [30].

## 3. Materials and Methods

### 3.1. Materials

PTFE (PN90, GaloPolymer, Russia) was used as the polymer matrix, and it had a particle size of 46–135 μm and a density of 2.16 g/cm^3^. Chemically modified discrete CFs Belum (Svetlogorsk Khimvolokno, Belarus) were used as the fibrous filler. The width and length of the CF were 8–10 μm and 50–500 μm, respectively. The CFs were modified by a plasma–chemical treatment in an environment with fluororganic compounds, based on a previous method [31]. Kaolin (Kl) with a chemical composition of Al_2_[Si_2_O_5_](OH)_4_ was used as a mineral filler, sourced from deposits from the Krasnoyarsk region in Russia. The average particle size of the Kl used in this study was 0.5–6.9 μm. The second natural filler used here was vermiculite (Vl), sourced from the Illino deposit, Sakha (Yakutia) Republic, Russia, with an average particle size of 0.5–7.5 μm. The chemical composition of Vl is (Mg^+2^, Fe^+2^, Fe^+3^)^3^[(Al, Si)_4_O_10_]·(OH)_2_·4H_2_O.

### 3.2. Sample Preparation

Mechanical activation of the layered silicates was carried out using a planetary mill (Activator-2S, Activator, Novosibirsk, Russia) for 2 min at 90 G. The test specimens were prepared according to a typical process: dry-mixing the polymer and filler in a paddle mixer, molding under a pressure of 50 MPa at room temperature for 2 min, then sintering the specimen in a programmable muffle furnace (SNOL 180/400) at 375 °C.

### 3.3. Characterization

The mechanical properties, such as the tensile strength, elongation at break, and yield strength, were determined according to the Russian standard GOST 11262-80 (plastics, tensile test method) using a universal testing machine (Autograph AGS-J, Shimadzu, Kyoto, Japan) at 25 °C, and a strain rate of 100 mm/min. The compressive stress at a deformation of 10% was measured following standard GOST 4651-2014 (plastics, compression test method, ISO 604: 2002, MOD) using the same instrument at a strain rate of 1 mm/min. The densities of the PTFE and PCM samples were determined according to the Russian standard GOST 15139-69 (plastics, methods for the determination of density). This method was developed for determining the bulk density of molded products (e.g., rods, bars, and tubes), and it provides a density measurement accuracy of 0.1%. Distilled water was used as the media.

The friction coefficients of PTFE and its PCM were determined according to the Russian standard GOST 11629-75 (plastics, method for testing the friction coefficient) using a tribometer (UMT-3, CETR, Mountain View, CA, USA). The pin-on-disc testing method was used for tribological characterization. The counterbody was a #45 carbon-steel disk with a hardness of 45–50 HRC, and a roughness R = 0.06–0.08 μm. We used a normal loading force of 160 N, an average sliding speed of 0.2 m/s, and a test duration of 3 hr. The test specimens were cylinders with diameters of 10 ± 0.1 mm, and heights of 20 ± 1 mm. The mass of the specimens was measured before and after sliding, using a high-precision analytical balance (OHAUS, Parsippany-Troy Hills, NJ, USA). The wear rate *K* (mm^3^/Nm) was estimated as follows:(2)$$K=\frac{V_{lost}}{\left(F_{N}d\right)}=\frac{m_{lost}}{\left(\rho F_{N}d\right)}$$ where *V_lost_* is the volume lost during sliding (mm^3^), *m**_lost_* is the mass lost during sliding (g), *ρ* is the density of the composite, *F_N_* is the normal force (N), and *d* is the sliding distance (m) [32].

The crystal structure of the PTFE and its PCMs was determined using X-ray powder diffractometry (XRD; ARL X'Tra, Thermo Fisher Scientific, Ecublens, Switzerland) with a CuK_α_ (λ = 0.154 nm) radiation source. The degree of crystallinity was estimated from the ratio of the areas of reflections corresponding to amorphous and crystalline regions. WinXRD software (v. 2.0-6, ThermoFisher, Ecublens, Switzerland) was used for data analysis.

The worn surface and supramolecular structures of the PTFE and PCM samples were characterized using scanning electron microscopy (SEM; JSM-7800F LV, JEOL, Tokyo, Japan). The specimens for supramolecular structure studies were obtained by cold chipping in liquid nitrogen. Fourier transform infrared spectroscopy (FT-IR; Varian 7000, Varian, Palo Alto, CA, USA) was used to record IR spectra with an attenuated total reflection (ATR) attachment over the range of 550–4000 cm^−1^. IR spectra were obtained before and after friction tests.

## 4. Conclusions

The use of a mixed filler containing mechanically activated layered silicates and CF in PTFE composites resulted in a significant increase in wear resistance, and improved mechanical properties. The improved mechanical properties were attributed to good intermolecular interaction between CF and PTFE, as a result of the plasma–chemical modification of the CF surface. The improved tribological properties of PCMs with mixed fillers were due to the formation of tribofilms containing layered silicates and PTFE fragments. Moreover, plasma–chemical modification of the CF surface can contribute to the formation of a strong tribofilm, due to increased intermolecular interaction between CFs and wear debris. The combination of two different fillers resulted in PCMs with properties that could not be achieved using a single filler. Further studies of the interactions between different kinds of fillers in polymer matrices are required, in order to develop new materials with unique characteristics.

## Figures and Tables

**Figure 1 molecules-24-00224-f001:**
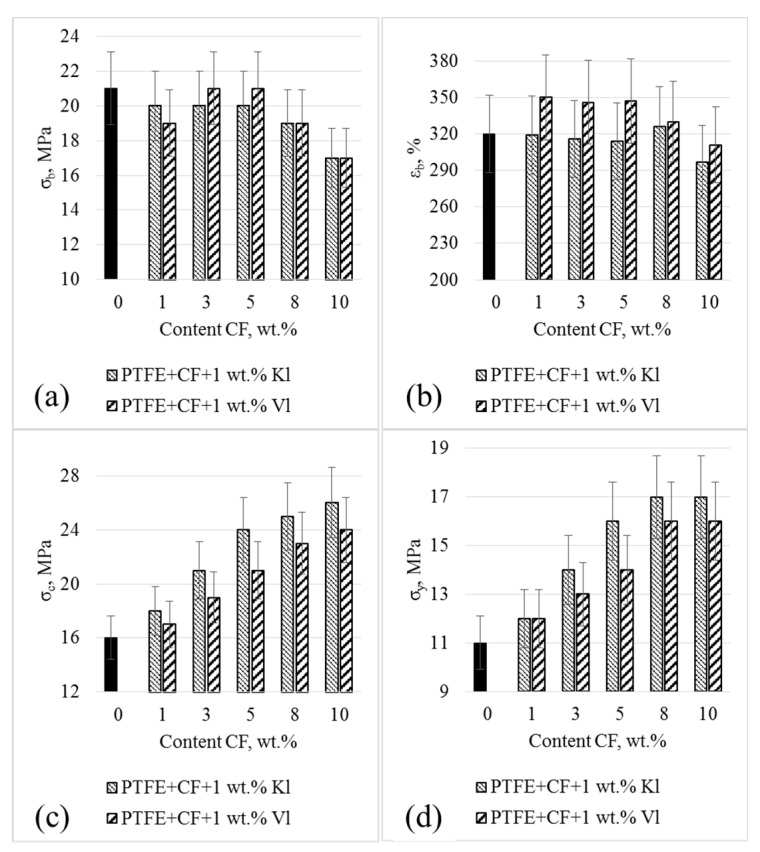
Tensile strength (**a**), elongation at break (**b**), compressive strength at 10% strain (**c**) and yield strengths (**d**) of polytetrafluoroethylene (PTFE) and polymer composite materials (PCMs).

**Figure 2 molecules-24-00224-f002:**
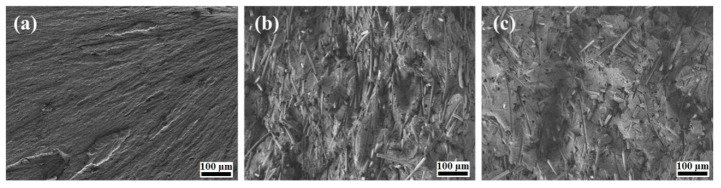
Scanning electron microscopy (SEM) image of supramolecular structure of PCM (×150): (**a**) initial PTFE; (**b**) PTFE + 8 wt % CF + 1 wt % Kl; (c) PTFE + 8 wt % CF + 1 wt % Vl.

**Figure 3 molecules-24-00224-f003:**
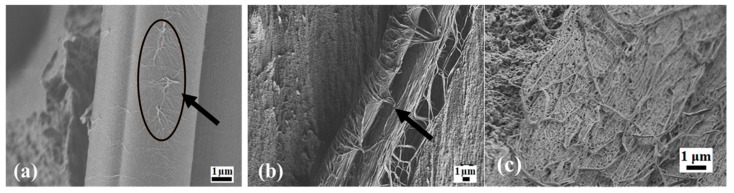
SEM image of the CF surface at different magnifications: (**a**) PTFE + CF + Vl × 10,000; (**b**) PTFE + CF + Kl × 3,000; (**c**) PTFE + CF × 10,000.

**Figure 4 molecules-24-00224-f004:**
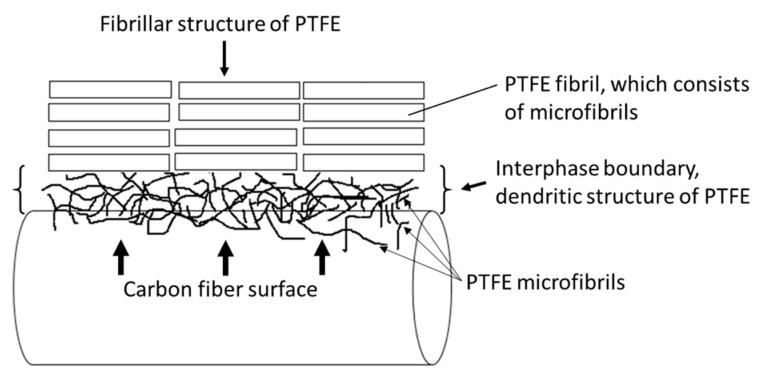
Scheme of the supramolecular structure at the boundary layer between PTFE and CF.

**Figure 5 molecules-24-00224-f005:**
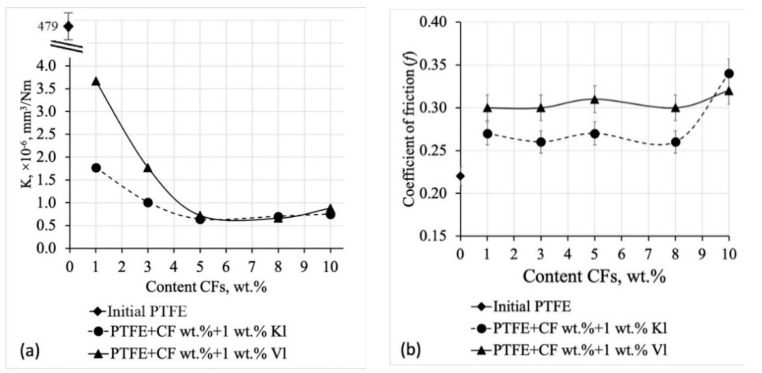
Specific wear rate (**a**) and friction coefficient (**b**) of PTFE and PCM.

**Figure 6 molecules-24-00224-f006:**

SEM image of PCM worn surfaces (×150): (**a**) initial PTFE; (**b**) PTFE + 5 wt % CF + 1 wt % Kl; (**c**) PTFE + 5 wt % CF + 1 wt % Vl.

**Figure 7 molecules-24-00224-f007:**
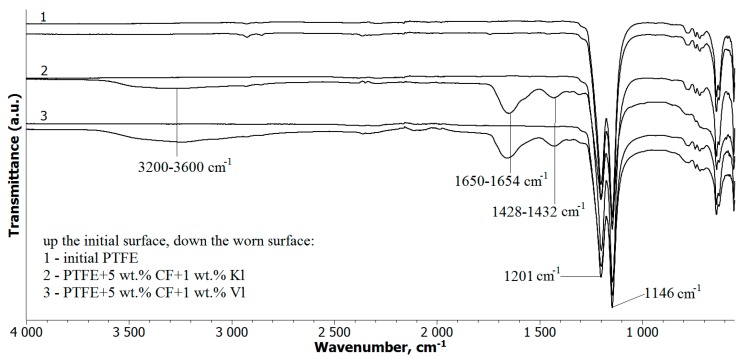
IR spectra of before and after the friction test of PTFE and PCM.

**Figure 8 molecules-24-00224-f008:**
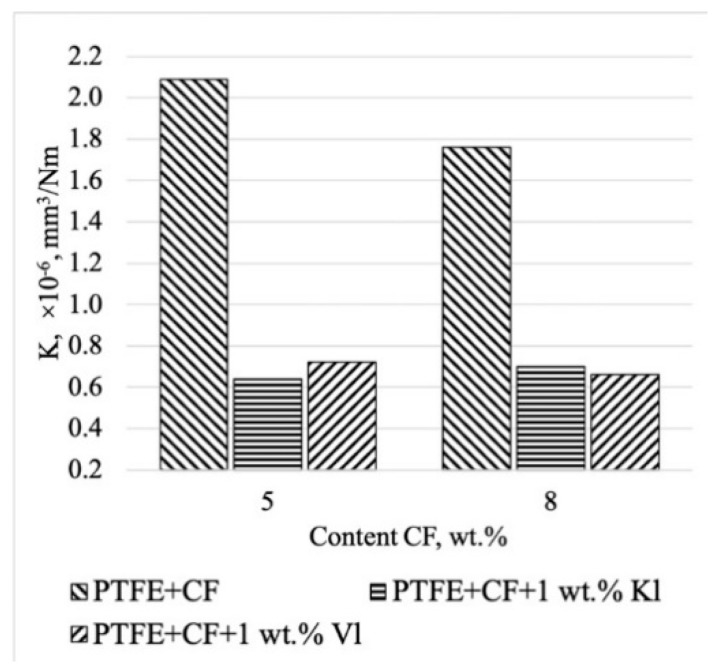
Comparison of the wear rate of the PCM complex filling and PCM-only carbon fibers.

**Figure 9 molecules-24-00224-f009:**
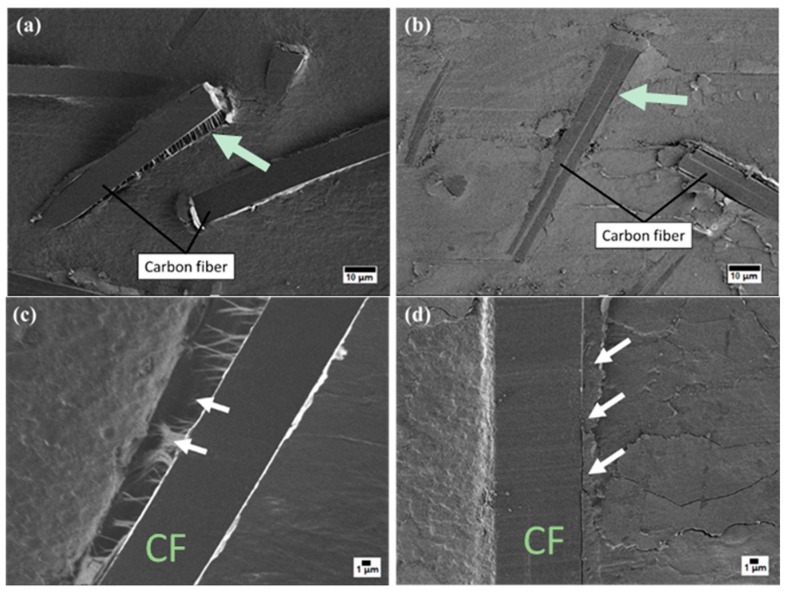
SEM image of PCM worn surfaces: (**a**) PTFE/CF (×1000); (**b**) PTFE/CF/Kl (×1000); (**c**) PTFE/CF (×3000); (**d**) PTFE/CF/Vl (×3000).

**Figure 10 molecules-24-00224-f010:**
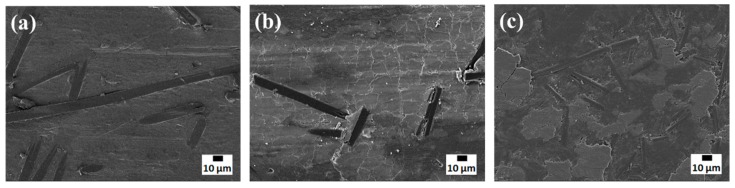
SEM micrographs of worn surfaces of the PTFE/CF (**a**), PTFE/CF/Vl (**b**) and PTFE/CF/Kl (**c**).

**Table 1 molecules-24-00224-t001:** Theoretical and experimental density and a crystallinity degree of PCM.

Content of CF, wt %	Theoretical Density, g/cm^3^	Experimental Density, g/cm^3^					XRD, %
	PTFE/CF	PTFE/CF/Kl	PTFE/CF/Vl	PTFE/CF	PTFE/CF	PTFE/CF/Kl	PTFE/CF/Vl
0	2.20	2.15 ± 0.01	2.16 ± 0.01	2.16 ± 0.01	63 ± 1	65 ± 1	64 ± 1
1	2.19	2.15 ± 0.01	2.16 ± 0.01	2.15 ± 0.01	64 ± 1	67 ± 1	66 ± 1
3	2.18	2.13 ± 0.01	2.14 ± 0.01	2.13 ± 0.01	68 ± 1	69 ± 1	69 ± 1
5	2.16	2.11 ± 0.01	2.11 ± 0.01	2.10 ± 0.01	69 ± 1	72 ± 1	71 ± 1
8	2.13	2.09 ± 0.01	2.08 ± 0.01	2.06 ± 0.01	71 ± 1	74 ± 1	73 ± 1
10	2.11	2.06 ± 0.01	2.06 ± 0.01	2.03 ± 0.01	73 ± 1	74 ± 1	74 ± 1

Note: CF—carbon fiber; Kl—kaolinite; Vl—vermiculite; Vl and Kl contents are 1 wt %.

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
