# Peer review of "Mechanical and Tribological Properties of Polytetrafluoroethylene Composites with Carbon Fiber and Layered Silicate Fillers"

_molecules, 2019, doi:10.3390/molecules24020224_

Round 1
Reviewer 1 Report
Well written form beginning to end.
The authors have investigated mixtures of silicates (vermiculite and kaolinite) and carbon fibers as fillers for PTFE. They found an increase in yield strength and compressive strength of the polymer when a mixed filler was used. In addition, the wear resistance increased. The paper is well structured. Relevant literature is reviewed and discussed. Materials and experimental methods are well described. Results are presented using diagrams and SEM pictures. Results are discussed and reasons for increase in material properties using fillers are discussed. Finally, conclusions are drawn from the experimental studies. Outline, content and English language are in perfect conditions. I recommend acceptation of the manuscript.
Author Response
For Reviewer 1
Thank you for your consideration.
With best regards
Reviewer 2 Report
The list of detail remarks is given below:
What was the shape of Kl and Vl particles?
In explanation of equation (1) there is a lack of density.
Page no. 5 - please change the russian letter in density of crystalline phase.
Please explain the mechanism of simultaneous better wear resistant and higher friction coefficient.
The values of wear rate in Fig. 8 are not comparable with the ones in Fig. 5, please comment it.
Author Response
For Reviewer 2
1) What was the shape of Kl and Vl particles?
The mechanically activated KL and Vl consisted of agglomerates of lamellar flakes.
2) In explanation of equation (1) there is a lack of density.
We appreciate the reviewer highlighting this oversight. We have changed the explanation of this equation.
3) Page no. 5 - please change the russian letter in density of crystalline phase.
This correction was made.
4) Please explain the mechanism of simultaneous better wear resistant and higher friction coefficient.
The tribofilm with high wear resistance and friction coefficient is formed on the composite surface during friction, due to changes in the structure and content of the worn surface of the composite. The composite fillers have a higher friction coefficient than PTFE, resulting in a tribofilm with a higher friction coefficient than the initial polymer. A better explanation of this has been added (page X, line Y).
5) The values of wear rate in Fig. 8 are not comparable with the ones in Fig. 5, please comment it.
We apologize for this confusing information. In the case of Fig. 5, we forgot to convert mass wear rate (mg/h) to wear rate (mm3/Nm). The data in Fig. 5 has now been corrected.
Reviewer 3 Report
The manuscript deals with the mechanical and tribological characterization of composite PTFE obtained by dispersion of mixed filler based on Carbon fibers and layered silicates. The use of hybrid CF-layered silicates it is here for the first time proposed. The results demonstrate that an overall improvement of the mechanical and tribological properties is achieved only if hybrid fillers are used. However, some critical aspects emerged:
1) Vermiculite and kaolinite strongly differs in terms of structure and intercalation properties, however the results obtained seem to be not affected from the filler type. How the authors can explain that?
2) The authors demonstrated that layered particles are primarily involved in the formation of a tribofilms, preventing the carbon fibers to be removed from the friction surface. Vermiculite and kaolinite are both double-layered materials. Do they keep their stacking in the composite or do they exfoliate? In my opinion XRD data can also be used to clarify this crucial point.
3) Lines 178-184: the discussion results to be not fluent.
4) Line 247: “.... detected by FT-IR spectroscopy (Fig. 6)”. FT-IR data are reported in Fig. 7
Author Response
For Reviewer 3
1) Vermiculite and kaolinite strongly differs in terms of structure and intercalation properties, however the results obtained seem to be not affected from the filler type. How the authors can explain that?
This is perhaps due to the fact that the mass content (1 wt.%) of the layered silicates in the composites was low compared to that of the CF. Therefore, the effect of the type of the layered silicate could have been diminished by the introduction of a large number of hydrocarbons.
2) The authors demonstrated that layered particles are primarily involved in the formation of a tribofilms, preventing the carbon fibers to be removed from the friction surface. Vermiculite and kaolinite are both double-layered materials. Do they keep their stacking in the composite or do they exfoliate? In my opinion XRD data can also be used to clarify this crucial point.
The high viscosity of PTFE (1011 Poise) prevents the exfoliation of silicates; moreover, there are no prerequisites for this. The authors conducted X-ray analysis, but due to the low content (1 wt.%) of layered silicates in the polymer composite, we could not observe any clear effect of exfoliation using the available equipment.
3) Lines 178-184: the discussion results to be not fluent.
This was part was edited in the revised manuscript, and we hope it is not easier to understand.
4) Line 247: “.... detected by FT-IR spectroscopy (Fig. 6)”. FT-IR data are reported in Fig. 7
This mistake was corrected in the revised article.
In addition, we changed Fig. 5 and 8. We replaced the commas with decimal points (for example 0,5 → 0.5).
Round 2
Reviewer 3 Report
The authors made the required changes.
In the revised manuscript the results are now well discussed and corroborate the conclusions of the authors.
I recommend the acceptation of the manuscript in the present form.